# Pertactin contributes to shedding and transmission of *Bordetella bronchiseptica*

Longhuan Ma[1]*, Kalyan K. Dewan[1], Dawn L. Taylor-Mulneix[1], Shannon M. Wagner[1], Bodo Linz[1], Israel Rivera[1], Yang Su[1,2], Amanda D. Caulfield[1], Uriel Blas-Machado[3], Eric T. Harvill[1]

**1** Department of Infectious Diseases, College of Veterinary Medicine, University of Georgia, Athens, Georgia, United States of America, **2** Department of Biochemistry, University of Georgia, Athens, Georgia, United States of America, **3** Department of Pathology, Athens Veterinary Diagnostic Laboratory, College of Veterinary Medicine, University of Georgia, Athens, Georgia, United States of America

* longhuanma@gmail.com

**Data Availability Statement:** All relevant data are within the manuscript and its Supporting information files.

## Abstract

Whooping cough is resurging in the United States despite high vaccine coverage. The rapid rise of *Bordetella pertussis* isolates lacking pertactin (PRN), a key vaccine antigen, has led to concerns about vaccine-driven evolution. Previous studies showed that pertactin can mediate binding to mammalian cells in vitro and act as an immunomodulatory factor in resisting neutrophil-mediated clearance. To further investigate the role of PRN *in vivo*, we examined the functions of pertactin in the context of a more naturally low dose inoculation experimental system using C3H/HeJ mice that is more sensitive to effects on colonization, growth and spread within the respiratory tract, as well as an experimental approach to measure shedding and transmission between hosts. A *B. bronchiseptica* pertactin deletion mutant was found to behave similarly to its wild-type (WT) parental strain in colonization of the nasal cavity, trachea, and lungs of mice. However, the pertactin-deficient strain was shed from the nares of mice in much lower numbers, resulting in a significantly lower rate of transmission between hosts. Histological examination of respiratory epithelia revealed that pertactin-deficient bacteria induced substantially less inflammation and mucus accumulation than the WT strain and *in vitro* assays verified the effect of PRN on the induction of TNF-α by murine macrophages. Interestingly, only WT *B. bronchiseptica* could be recovered from the spleen of infected mice and were further observed to be intracellular among isolated splenocytes, indicating that pertactin contributes to systemic dissemination involving intracellular survival. These results suggest that pertactin can mediate interactions with immune cells and augments inflammation that contributes to bacterial shedding and transmission between hosts. Understanding the relative contributions of various factors to inflammation, mucus production, shedding and transmission will guide novel strategies to interfere with the reemergence of pertussis.

**Funding:** This work was supported by grants 1R21DC018496-01A1 and 5R21AI142678-02 of the National Institutes of Health to ETH. The funders had no role in study design, data collection and analysis, decision to publish, or preparation of the manuscript.

## Author summary

*B. pertussis* strains lacking pertactin have been rising in prevalence especially in countries using acellular vaccines containing pertactin as a key, membrane-associated surface antigen. Previous *in vivo* studies revealed immunomodulatory properties of pertactin in conventional *B. pertussis* infection models in which roughly one million bacteria are delivered into lungs, leading to severe pneumonic disease and a strong immune response. However, natural infections begin in the nasopharyngeal region, progress slowly during a prolonged catarrhal stage, only later reaching the trachea and rarely involve the lungs. In this study, a more natural experimental system takes advantage of the ability of *B. bronchiseptica*, a closely related species, to naturally colonize mice with inocula as low as 5 colony forming units (CFU). In this system *B. bronchiseptica* can be observed to efficiently colonize, grow, spread within the respiratory tract, is shed from the nares, and transmits between hosts, allowing each of these steps to be measured and studied. Under these conditions, an isogenic pertactin deletion strain was indistinguishable from its parental strain in its abilities to colonize, grow in numbers and spread within the respiratory tract. However, the pertactin-deficient mutant was shed from these mice in lower numbers than wild type, and was defective in transmission between mice. These assays reveal novel roles of pertactin in the induction of inflammation, mucus production, shedding and transmission.

## Introduction

Pertussis, or whooping cough, is an acute respiratory disease caused by the gram-negative bacterial pathogen *Bordetella pertussis*. Historically, pertussis had a high morbidity and mortality rate and was the predominant childhood killer before the introduction of whole cell vaccines (WCV) that greatly reduced its prevalence [1]. In highly vaccinated populations, the reduced incidence of severe disease led to attention being focused on the moderate side effects associated with WCV, including common local reactions (swelling and pain), uncommon systemic reactions (fever, irritability, drowsiness, loss of appetite and vomiting), and very rare neurological reactions (acute encephalopathy in newborns) [2]. To address these concerns, less reactive pertussis acellular vaccines (ACV) containing up to five purified, detoxified *B. pertussis* proteins, including pertussis toxoid, pertactin, type 2 and 3 fimbriae, and filamentous hemagglutinin, were increasingly used in the 1980s and 1990s, and eventually replaced the WCV in most developed countries [3]. Coincident with the switch to ACV, the incidence of whooping cough has been increasing in all age groups, from infants to adults [4]. Moreover, in recent years several countries have seen an increased percentage of clinical isolates that fail to express pertactin (PRN), a prominent outer membrane protein [5,6]. The loss of PRN, a prominent ACV component, has led to concern that ACV vaccines are driving the evolution of *B. pertussis* to evade vaccine-induced immunity [7]. The long-term consequences of this possibility are difficult to predict as we have limited knowledge of the biological functions of PRN.

PRN is an autotransporter protein that appears on the outer membrane of *B. pertussis* [8]. This protein has been included in acellular vaccines in several countries, based on evidence that pertactin specific antibodies contribute to protection against pertussis. Previous studies revealed that PRN exerts cell binding through its RGD motif and contributes to adherence and invasion of mammalian cells [9–12]. However, the adhesion functions of pertactin were not detected in some other studies [13–17]. *In vivo* studies have shown an immunomodulatory role of PRN [17,18], in the context of severe pneumonic infection initiated by super-natural challenge with roughly a million CFU of *B. pertussis* or *B. bronchiseptica* delivered deep into

the lungs of mice where they caused severe lung pathology. This experimental system was established to study the most extreme form of disease and has enabled the development and testing of vaccines and therapeutics to prevent and treat severe disease in the lower respiratory tract. But *B. pertussis* is highly infectious, suggesting very small initial inocula, and natural infections begin with the weeks-long catarrhal stage, which is limited to the nasopharyngeal region and is a highly contagious period. Thus, colonization with low initial numbers followed by weeks of growth within and shedding from the upper respiratory tract are critical aspects of the infectious cycle of *B. pertussis*. However, these aspects are not simulated in the standard inoculation approach that delivers as many as a million bacteria deep into the lungs and measures the consequences of the subsequent extreme pneumonic pathology [19–21].

We have focused on simulating the progression of natural infections in order to study the contributions of individual bacterial factors and host immune functions [22–25]. We use *B. bronchiseptica* because it naturally and highly efficiently colonizes, grows, causes pathology, is shed, and transmits between mice. Importantly, it can efficiently colonize mice when delivered to the external nares in very small numbers (<5 CFU), allowing the entire natural progression of the infection to be studied, including many aspects that are obviated by the conventional approach of washing millions of bacteria deep into the lungs [22–25]. *B. pertussis* and *B. bronchiseptica* are very closely related and shared genes are ~ 98% identical at the nucleotide level, but differ in some notable regards. While most of the best studied factors are shared, each expresses a somewhat different subset of factors [5,26–31], that are likely to explain their differences in host range and/or the diseases they cause: while *B. pertussis* cause an acute and severe disease in humans, *B. bronchiseptica* causes infection that can range in severity and often persists in the upper respiratory tract of infected animals. However, similarities far outbalance differences. Both species infect mammals and, with the notable exception of expression of the pertussis toxin, share nearly all genes implicated in interacting with their hosts, including the vaccine antigens fimbriae (FIM), filamentous hemagglutinin (FHA) and both species express PRN orthologs that are 92% similarity of PRN at the amino acid level and are likely to perform similar functions [31].

The study of the true in vivo functions of any of *Bordetella* spp. factors requires experimental systems in which their effects can be measured. Unfortunately, *B. pertussis* poorly colonizes mice, making studies of the details of natural infections challenging. Unlike *B. pertussis*, *B. bronchiseptica* efficiently colonizes mice with an infectious dose less than 5 CFU with infections progressing efficiently, allowing the study of the mechanistic basis for their complex interactions in the context of naturally progressing infection in the natural host [22–25]. We also previously described *B. bronchiseptica* efficiently transmitting among TLR4-deficient C3H/HeJ mice. *B. bronchiseptica* grows and is shed in higher numbers in these mice, allowing more efficient transmission [22–25]. This experimental model allows all the bacterial components necessary for efficient transmission to be experimentally manipulated to better understand bacterial mechanisms involved in the transmission process.

In this study, to investigate the biological role(s) of PRN in the infectious process and pathogenicity of *Bordetella* species, we used a mutant of *B. bronchiseptica* with an in-frame deletion in the *prn* gene which was generated by allelic exchange as described by Inatsuka *et al.* [17]. *In vivo* experiments showed that *prn* was not necessary for efficient colonization or early growth in the nasopharynx, but was required for efficient bacterial shedding and transmission between mice. Furthermore, wild-type *B. bronchiseptica* induced higher levels of inflammation and more mucus secretion than the Δ*prn* mutant, revealing a role for PRN in promoting inflammation and mucus secretion, which was supported by *in vitro* assays showing higher secretion levels of the pro-inflammatory cytokine TNF-α from WT infected compared to Δ*prn* mutant infected murine macrophages. Moreover, the WT strain, but not the *prn* deletion

strain, was recovered from splenocytes of infected mice, indicating that PRN contributes to to systemic dissemination that involves intracellular survival. Together these data suggest that PRN promotes the induction of inflammation and mucus production, mediates shedding and transmission to new hosts and is involved in the intracellular survival and systemic dissemination of the pathogen.

## Results

### PRN is not necessary for *B. bronchiseptica* to efficiently colonize the host and grow within the respiratory tract

To evaluate the contribution of pertactin to various aspects of the biology of *B. bronchiseptica*, we used an isogenic *prn* gene deficient mutant (*BbΔprn*) of *B. bronchiseptica* [17]. The parental strain RB50 (*Bb* WT) is well-established as being highly efficient in colonizing, persisting, and transmitting among mice [22–25]. To confirm the deletion was clean and not complicated by other changes, we re-sequenced the whole genome and confirmed the in-frame deletion of *prn* (S1 Fig). The *prn* mutant strain showed no defect in *in vitro* growth, adherence to human alveolar epithelial cells or cytotoxicity to murine RAW 264.7 macrophages (S2–S4 Figs). Using the conventional pneumonic infection model, C57BL/6 mice were inoculated intranasally with 50 μL PBS buffer containing $5x10^5$ CFU of either WT or *BbΔprn* bacteria. Bacterial numbers in nasal cavities, trachea and lungs harvested at 7-, 14- and 28-days post-inoculation (dpi) did not differ significantly between mice infected with *Bb* WT or *BbΔprn*, (S5 Fig). Interestingly, *BbΔprn* showed higher colonization levels in the nasal cavity and lungs at 14 dpi, but these differences were not observed in later timepoints. Histopathological analysis of nasal cavities at 7 dpi and 14 dpi detected mild suppurative inflammation in both groups with similar incidence and severity. Thus, the standard high dose pneumonic infection experimental system did not reveal a significant role for PRN in the pathogenesis of *B. bronchiseptica*. Rather than extending the use of this conventional experimental system that generates extreme lung pathology but poorly simulates natural infection, we considered newer assays and approaches we have developed.

### PRN contributes to transmission

We recently developed an experimental system to study various aspects of the process of transmission between hosts using C3H/HeJ mice [22–25,32]. To relate shedding and transmission to other aspects of the infectious process often assumed to affect shedding, such as bacterial load, we first assessed the time course of infection in these mice. C3H/HeJ mice were inoculated with 150 CFU of either *Bb* WT or *BbΔprn* in 5 μL of PBS, a volume that deposits the bacterial inoculum only into the nasal cavity. Respiratory organs were harvested at 7-, 14- and 28-dpi. Both bacteria grew over time, with similar numbers observed in nasal cavity, trachea and lung (S6 Fig), suggesting that PRN has no critical role in colonization, growth, and spread within the respiratory tracts of these mice.

To assess the contribution of PRN to transmission, pairs of C3H/HeJ mice were inoculated with 150 CFU of either *Bb* WT or *BbΔprn* strain (in index mice) and then co-housed with two naïve C3H/HeJ mice (uninfected recipient mice) in cages of four. To evaluate bacterial transmission, the nasal cavities of 12 such co-housed naïve mice per strain (from 6 cages) were assessed for bacterial colonization after 21 days of co-housing with infected mice. In the *Bb* WT group, all 12 co-housed naïve mice became colonized, while *BbΔprn* bacteria were only transmitted to 5 out of 12 co-housed naïve mice (Fig 1A), suggesting that PRN contributes to bacterial transmission between hosts.

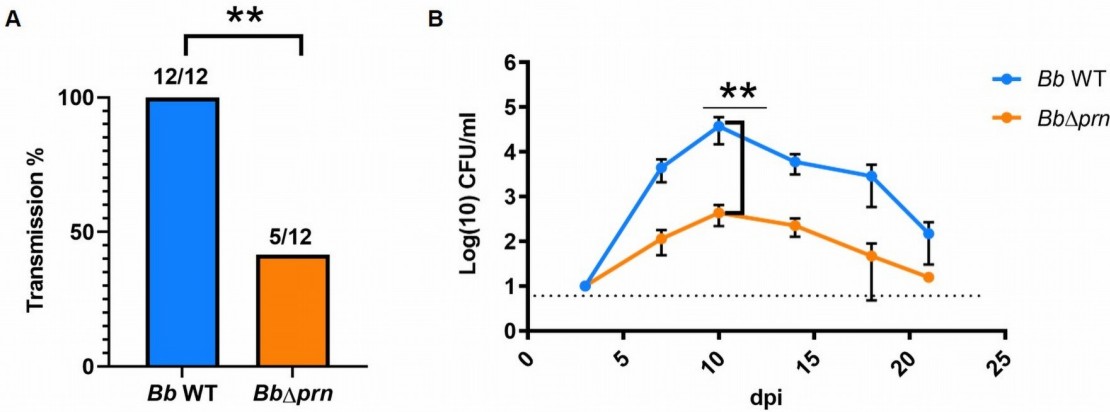

**Fig 1. *B. bronchiseptica* PRN contributes to bacterial transmission and shedding.** A) Transmission rates among *Bb* WT (blue) or *BbΔprn* (orange) inoculated mice. B) Number of bacteria shed from the external nares of *Bb* WT infected mice (blue) and *BbΔprn* infected (orange) mice. Twelve mice were utilized in each time point per group. The dashed lines show the detection limit in the experiment. Error bars show the standard error of mean. Statistical significance was calculated by using chi-square test in panel A and two-way ANOVA in panel B. **p < 0.01.

## PRN contributes to bacterial shedding

The deficiency of *BbΔprn* in transmission could be due to either a defect in shedding from infected hosts or colonization of exposed hosts. To test the efficiency of colonization, we inoculated C3H/HeJ mice with decreasing doses of bacteria. Both *Bb* WT and *BbΔprn* efficiently colonized the respiratory tracts of all mice with a calculated inoculation dose of ~5 CFU (S7 Fig) indicating that PRN is not required for efficient colonization of the respiratory tract.

Since PRN contributes to transmission but is not required for efficient colonization, we investigated its effect on shedding from the noses of challenged mice. C3H/HeJ mice (n = 12) were inoculated with 150 CFU of either *Bb* WT or *BbΔprn*, and bacteria shed from the nose were collected by gently swabbing the external nares with Dacron-polyester tipped swab every two- or three-days post inoculation. Throughout the experiment, WT bacteria were shed from the nares at high numbers, > 1000 CFU at multiple consecutive timepoints, indicating prolonged, high level shedding. In contrast, the *BbΔprn* strain was shed at 10 to 100-fold lower numbers from the first day of shedding to the end of the experiment at day 21 (Fig 1B), revealing a critical role for PRN in efficient shedding from the nose to the environment.

## PRN contributes to inflammation in the nasal cavity

The striking difference in shedding, without a substantially different load of bacteria in the respiratory tract, led us to speculate that PRN might induce shedding by altering the inflammatory state and/or mucus production. To test this hypothesis, C3H/HeJ mice infected with *Bb* WT or *BbΔprn* were collected at 7 and 14 dpi for histopathological analysis. Histopathology showed that in the WT group, all 10 analyzed mice had inflammation in the nasal cavity ranging from mild to severe (Fig 2G–I). In contrast, of the *BbΔprn* infection group, 2 out of 10 had no apparent inflammation and the other 8 had severity scores which varied from minimal to mild (Fig 2D–F). These data indicate that PRN is involved in induction of inflammation in the nasopharynx (Fig 3A and S1 Table). To probe the effect of PRN on immune cell recruitment, flow cytometry was used to analyze immune cell populations in nasal cavities 14 days after inoculation with *Bb* WT or *BbΔprn* (Fig 4). *Bb* WT induced a significant increase in numbers

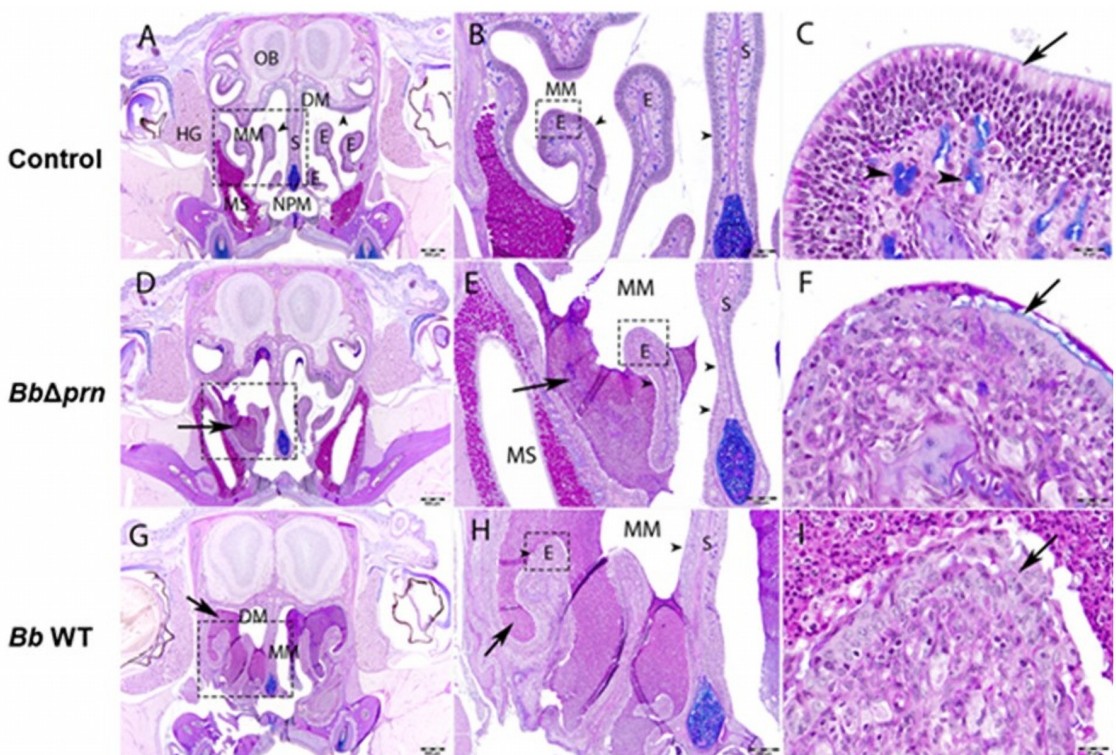

**Fig 2. *B. bronchiseptica* PRN induces inflammation and mucus secretion.** (A-I) Coronal sections of the nasal cavity (level 4) of control or *B. bronchiseptica (BbΔprn* or *Bb* WT strains) infected C57BL/6J mice at 14 dpi with stained with Alcian blue–Periodic acid-Schiff hematoxylin (AB-PASH) stain. A) Image of normal nasal cavity, level 4, from a PBS-inoculated control mouse. At this level, a layer of olfactory epithelium (arrowheads) lines the nasal mucosa within the dorsal and middle nasal meatus. Scale bar = 500μm. B) Higher magnification of Fig 2A (dashed rectangle) showing ethmoturbinates (E) and septum (S) lined by a layer of olfactory epithelium (arrowhead) within the middle meatus (MM). Scale bar = 200 μm. C) Higher magnification of Fig 2B (dashed rectangle) showing an ethmoturbinates lined by olfactory epithelium (arrow). Arrowhead points to a mucus producing (alcian blue-positive) gland within the lamina propria. Scale bar = 20 μm. D) Image of the nasal cavity, level 4, from a mouse inoculated with *BbΔprn*. The arrow points to a moderate accumulation of mucopurulent within the middle meatus. Scale bar = 500 μm. E) Higher magnification of Fig 2D (dashed rectangle) showing deposition of mucopurulent exudate (arrow) covering the olfactory epithelium (arrowhead) within the middle meatus (MM). There is thinning of the olfactory epithelium (arrowheads) along the septum (S) and ethmoturbinate (E). Scale bar = 200 μm. F) Higher magnification of Fig 2E (dashed rectangle) showing an ethmoturbinate. There is loss of olfactory epithelium and replacement by ciliated epithelium covered by a thin layer of mucus (arrow). There is loss of mucus producing glands within the lamina propria. Scale bar = 20 μm. G) Image of the nasal cavity, level 4, from a *B. bronchiseptica (RB50)*-inoculated mouse. The arrow points to a severe accumulation of mucopurulent within the dorsal (DM) and middle meatus (MM). Scale bar = 500 μm. H) Higher magnification of Fig 2G (dashed rectangle) showing deposition of mucopurulent exudate (arrow) covering the olfactory epithelium within the middle meatus (MM). There is thinning and loss of the olfactory epithelium (arrowheads) along the septum (S) and ethmoturbinate (E). Scale bar = 200 μm. I) Higher magnification of Fig 2H (dashed rectangle) showing an ethmoturbinate. Large numbers of PAS-positive neutrophils cover the ethmoturbinate. There is total loss of olfactory epithelium and replacement by non-ciliated epithelium (arrow). There is loss of mucus producing glands within the lamina propria. Scale bar = 20 μm. Abbreviations: Dorsal Meatus (DM); Ethmoturbinate (E); Harderian Gland (HG); Maxillary Sinus (MS); Middle Meatus (MM); Nasopharyngeal Meatus (NPM); Olfactory Bulb (OB); Septum (S). There were 5 mice in each time point per group.

of neutrophils, NK cells, macrophages, and B cells in the nasal cavities (Fig 4C–F). In contrast, *BbΔprn* induced an increase in macrophages, but no significant increase of other cell types compared to the uninfected control (Fig 4). These results indicate that PRN plays a role(s) in activation and recruitment of neutrophils, NK cells, and B cells, leading to higher inflammation levels in the nasal cavity.

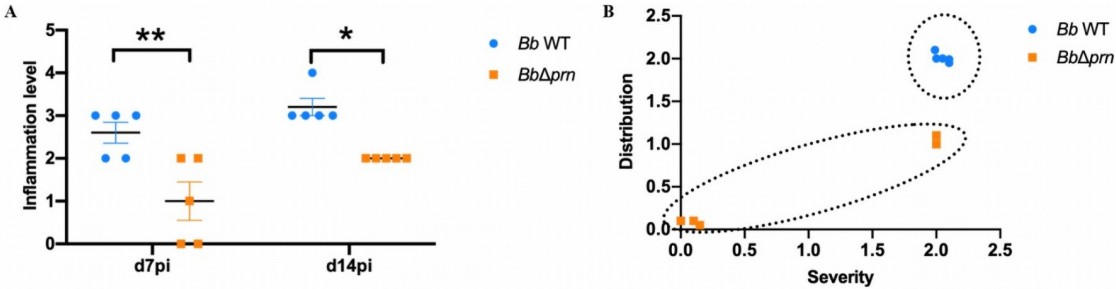

**Fig 3. PRN induces inflammation in the nasal cavity.** A) Inflammation level in the nasal cavities of mice inoculated with either *Bb* WT (blue) or *Bb* Δ*prn* (orange) at 7-and 14-dpi. B) Mucus accumulation in the nasal cavity of *Bb* WT (blue) or *Bb* Δ*prn* (orange) inoculated mice at day 7 pi. Individual samples from either group are highlighted with dashed circles. The veterinary pathologist conducting the inflammation level analysis (UBM) was blinded to the sample source. There were 5 mice in each time point per group. Error bar shows the standard error of mean. Statistical significance was calculated using Two-way ANOVA. $^*$p < 0.05, $^{**}$p < 0.01, $^{***}$p < 0.001.

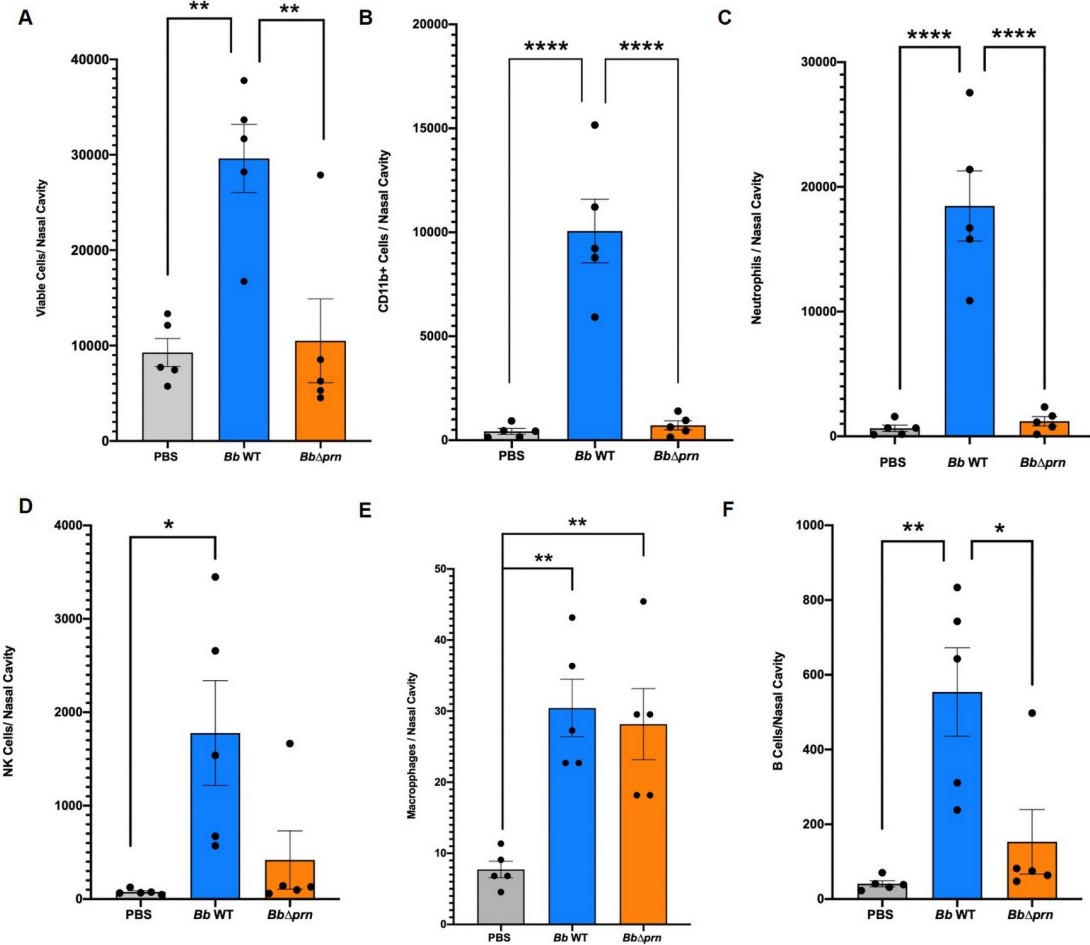

**Fig 4. *B. bronchiseptica* PRN contributes to the recruitment of leukocytes, neutrophils and B cells in the nasal cavity.** A) Total cells recruited in nasal cavities. B) Comparison of CD11b+ cells recruited in nasal cavities. C) Neutrophils recruited in nasal cavities. D) NK cells recruited in nasal cavities. E) Macrophages recruited in nasal cavities. F) B cells recruited in nasal cavities. There were 5 mice in each time point per group. Error bar shows the standard error of mean. Statistical significance was calculated using One-way ANOVA. $^*$p < 0.05, $^{**}$p < 0.01, $^{***}$p < 0.001, $^{****}$p < 0.0001.

## PRN increases mucus secretion in acute inflammation

The dramatically higher number of bacteria shed from mice infected with *Bb* WT than those given *BbΔprn* led us to examine whether increased mucus production might be involved. To compare mucus secretion in the nasal cavity of *Bb* WT infected and *BbΔprn* infected mice, Alcian blue-Periodic acid Schiff (PAS) staining was performed on tissue sections. At 7 dpi all 5 *Bb* WT infected mice displayed moderate mucus secretion. In contrast, only 2 out of 5 *BbΔprn* infected mice displayed some mucus accumulation and only in a small area, while the other 3 showed no mucus accumulation at all (Fig 3B). The substantially higher mucus secretion in *Bb* WT infected mice compared to *BbΔprn* infected mice indicates that PRN plays a role in the induction of mucus production during the acute phase of infection.

## PRN induces secretion of pro-inflammatory cytokines

Because of the very different numbers of neutrophils we observed in the noses of the two groups of mice, we hypothesized that PRN may affect the induction of two pro-inflammatory cytokines that recruit and activate neutrophils, IL-1 and TNF-α [33–35]. We harvested the nasal cavities of mice inoculated with *Bb* WT or *BbΔprn* at 24- and 48- hpi and observed somewhat higher levels of IL-1β and TNF-α in the former, albeit without statistical significance (S8 Fig). To test the effect of PRN on cytokine secretion from macrophages, we exposed macrophages to *Bb* WT or *BbΔprn* and observed higher levels of TNF- α in the former group, suggesting that PRN promotes secretion of TNF-α, but did not affect IL-1β (Fig 5 and S9 Fig).

## PRN contributes to intracellular survival and systemic dissemination

*Bb* has been shown to be able to invade and persist in immune cells, which can have profound effects on the local and systemic immune responses [36–38]. Since PRN has been implicated in interactions with immune cells, we tested its effects on access to and invasion of immune organs [17,39]. We harvested spleens from mice inoculated with 150 CFU of either *Bb* WT or *BbΔprn* at 7-, 14- and 28-dpi. *Bb* WT was recovered from 3 of 4 spleens and 5 of 8 spleens of

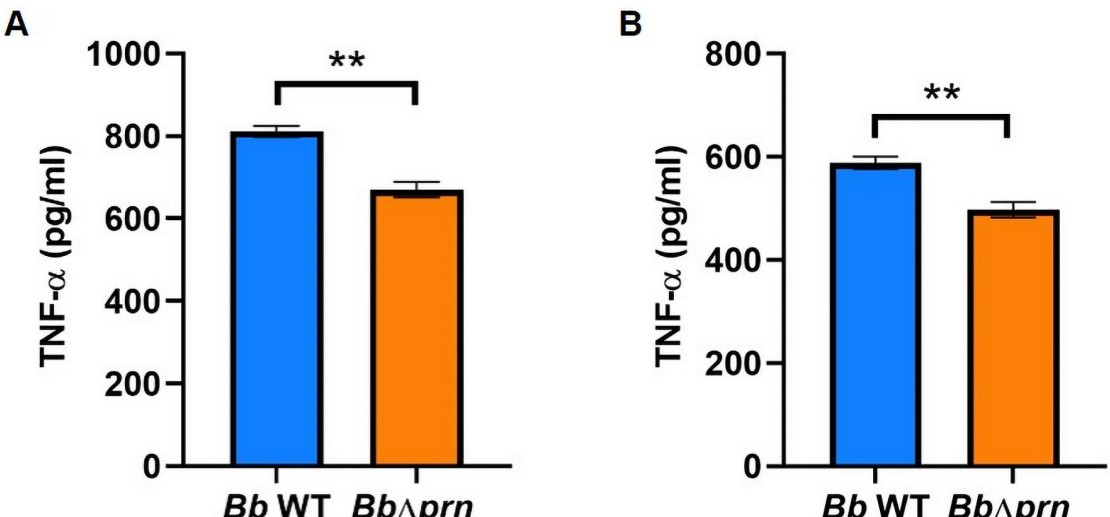

**Fig 5. PRN induces the secretion of TNF-α from RAW 264.7 macrophages.** TNF-α secretion from macrophages when exposed with *Bb* WT (blue) or *BbΔprn* (orange) for 1 hour with MOI at 10 (A) or MOI at 100 (B). Error bar shows the standard error of mean. Statistical significance was calculated using unpaired T test. *p < 0.05, **p < 0.01, ***p < 0.001.

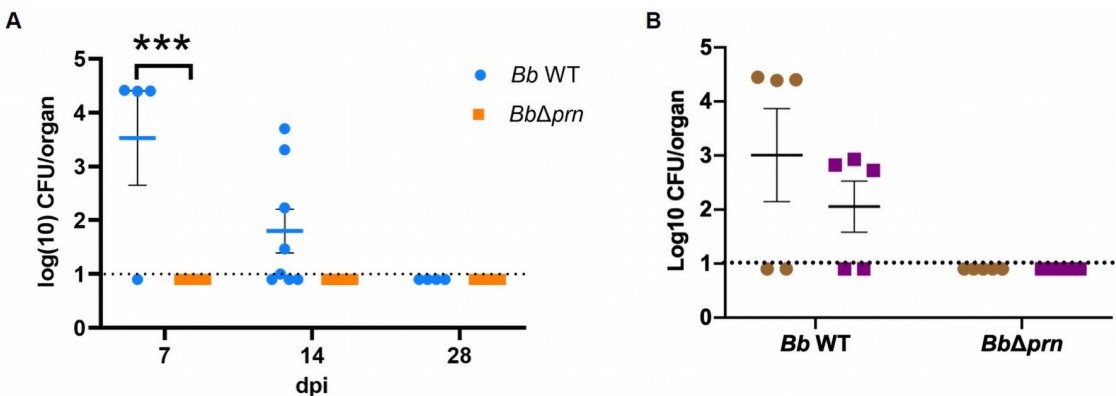

**Fig 6. PRN contribute to systemic dissemination and intracellular survival of *B. bronchiseptica*.** A) *Bb* WT were recovered from spleen of infected mice at 7- and 14 dpi, while there was no bacterial recovery from spleen in *BbΔprn*-infected mice in both time points. There were 4 mice for assays performed 7- and 28 dpi, and 8 mice for assays performed at 14 dpi per group. B) *Bb* WT were recovered from isolated splenocytes of infected mice at 7 dpi. The brown circles represent bacteria recovered from isolated splenocytes without gentamicin treatment while purple squares represent bacteria recovered from splenocytes post gentamicin treatment. The dashed lines show the detection limit in both experiments. Error bar shows the standard error of mean. Statistical significance was calculated by using Two-way ANOVA. *p < 0.05, **p < 0.01, ***p < 0.001.

infected mice on days 7 and 14, respectively, while no bacteria were found in the spleens of any *BbΔprn* inoculated mice (Fig 6A). To examine whether *Bb* WT in the spleen were surviving intracellularly, single cell suspension of splenocytes were treated with gentamicin treatment to kill the extracellular bacteria. Gentamicin killed 100% of control bacteria from culture (without host cells present), but did not kill all of those recovered from spleen, indicating that some were within host cells (Fig 6B), indicating that *Bb* WT expressing PRN can survive within the splenocytes. Additionally, we detected a higher IgG titer in *Bb* WT infected mice compared to that in *BbΔprn* infected mice (S10 Fig), suggesting that a strong immune response was induced to against this systemically disseminated infection.

## Discussion

Previous studies have used different and unrelated clinical isolates of *B. pertussis* with or without PRN, demonstrating, for example, that an isolate lacking PRN sustained a longer infection in mice immunized with an acellular pertussis vaccine [40]. Such differences cannot be specifically attributed to the loss of PRN, since the genomes of these two separate isolates differ in many other genes. Moreover, because *B. pertussis* poorly colonizes, sheds, and transmits between mice, it has been difficult to assess the role of PRN in these critical aspects of the infectious cycle. To address these problems, we and Inatsuka *et al*. used an in-frame deletion of *prn* in *B. bronchiseptica*, a close evolutionary progenitor of *B. pertussis* that naturally infects mice, to test the roles of PRN in infections of a natural host. In the conventional *Bordetella* spp. experimental system in which a very high inoculation dose (5 x $10^5$ CFU) delivered in a large volume (50 μL) to deposit bacteria deep in the lungs, PRN had a modest effect in BALB/c mice in the prior study [17]. In our high dose pneumonic challenge experiment, using different mice (C57BL/6) assessed at different time points, we did not observe measurable defects of the *prn* mutant. Since BALB/c and C57BL/6 mice have different immune responses to *B. pertussis* infection and vaccination [41–43], the different genetic backgrounds of two mouse strains and/or the different time points may explain the different observations in these two studies.

Rather those extensive comparisons between experimental conditions using the conventional experimental system, which is an extreme model of severe pneumonic disease, we directed our efforts toward assays that are more sensitive to various aspects of the infection and transmission processes. To better simulate a more natural, gradually progressing *Bordetella spp.* infection, we used a low dose inoculation model in which an experimental dose as low as 5 CFU is delivered into the external nares of mice in a 5μl droplet that deposits bacteria only in the nose. In this more natural experimental system, PRN was not required for efficient colonization and spread within inoculated hosts. When delivered only into the nasal cavity, rather than deep into the lungs, *Bb* WT-inoculated mice shed significantly more bacteria and transmitted between mice much more efficiently than *BbΔprn*-inoculated mice (Fig 1). *Bb* WT also induced significantly more inflammation and mucus production than RB50Δ*prn*. Inflammation and mucus production contribute to the rhinorrhea that facilitates shedding and transmission, as noted previously with *Streptococcus pneumonia* transmission [44]. The significantly higher number of neutrophils were observed in the nares of *Bb* WT infected mice, suggesting that PRN may play critical roles in recruitment and activation of neutrophils and/ or in resistance to neutrophils-mediated clearance, as was observed by Inatsuka CS et al [17]. PRN induced secretion of TNF-α from RAW 264.7 macrophages in *in-vitro*, which may explain the higher inflammation levels observed in mice infected with *Bb* WT. The role of TNF-α in the transmission of *B. bronchiseptica* deserves more attention in future studies as this factor may work as a potential therapeutic target for blocking transmission of *Bordetella spp.* and other pathogens.

PRN, as an outer surface protein with a demonstrated ability to affect adhesion to mammalian cells *in vitro* [9–11], is generally considered an "adhesin". However, it is not clear that PRN contributes to adhesion to either bronchial or laryngeal cells *in vivo*. In this study, deleting PRN did not affect the ability of *B. bronchiseptica* to efficiently colonize mice with a remarkably small inoculum, indicating that PRN is not required for the efficient adherence involved in initial colonization. However, *Bb* WT showed significantly higher recovery from spleens of infected hosts, indicating that PRN affects the ability to get to and/or survive in the spleen. The observation of substantial numbers of *Bb* WT, but not *BbΔprn*, within splenocytes (Fig 6), suggests PRN might affect invasion of immune cells and/or phagocytic killing within splenocytes. Upon encountering invading bacteria, immune cells trigger cascades of inflammatory responses by secreting cytokines, chemokines, small lipid mediators (SLM), and antimicrobial peptides (AMPs) that can contribute to increased phagocytic capacity and bacterial clearance [45], amongst other potential effects. After phagocytosis, peripheral immune cells can carry engulfed bacteria to deeper tissues including draining lymph nodes for T cell priming [46]. Thus, systemic dissemination of *Bb* WT, but not *BbΔprn*, may result in a stronger immune response against the former, consistent with the histopathology analysis and IgG antibody titers observed (Fig 2 and S10 Fig). In addition, the role of PRN in inducing TNF-α secretion may also contribute to the higher inflammation observed in the nasal cavity of *Bb* WT infected mice (Fig 5).

The increasing prevalence of PRN-deficient *B. pertussis* strains raises questions about both positive and negative (purifying) selection. It appears increasingly likely that PRN-containing ACVs select for PRN-deficient *B. pertussis*. It is also likely that the epidemiology of pertussis has changed as host behavior, population density, and worldwide travel affect the network of connected hosts. Even though PRN-deficient *B. pertussis* isolates may be shed less, a dense and well-connected population of vaccinated hosts in countries like the USA may be sufficient for a successful chain of transmission. Alternatively, the loss of PRN may have less cost in *B. pertussis* due to compensation by other genes. There are 15 other autotransporter genes in the genome of *B. pertussis* [5,31], one or more of which may compensate for the loss of PRN. Since

PRN plays roles in systemic dissemination and induction of inflammation in murine transmission models (Figs 2–6), other factors facilitating systemic dissemination and promoting inflammation may compensate for its loss. Alternatively, C3H/HeJ mice may fail to mimic immunocompetent hosts in every aspect of pathogenesis or *B. pertussis* may be different from *B. bronchiseptica* in various aspects, and it is possible that the functions of PRN in these two species may differ to some extent. However, to test this, an efficient *B. pertussis* transmission model is needed. In addition, further studies are needed to identify factors that might compensate for the loss of PRN in current circulating strains. This might lead to the identification of alternative antigens that could be included in next generation vaccines against *B. pertussis*.

# Material and methods

## Ethics statement

Mouse experiments used in this study were performed in strict accordance to recommendations outlined in the Guide for Care and Use of Laboratory Animals of the National Institute of Health. Protocols were approved by the Institutional Animal Care and Use Committee at University of Georgia (A2016 02-010-Y2-A3, *Bordetella*-Host Interactions). Mice were closely monitored during experiments and any mouse found moribund was euthanized using $CO_2$ inhalation to prevent unnecessary suffering.

## Bacterial strains and growth

*B. bronchiseptica* WT strain RB50 and isogenic pertactin knockout strain *Bb*Δ*prn* (SP5) have been previously described [11,17]. To allow mutant and WT to be distinguished in "competition assays", SM10λpir cells carrying allelic exchange vector pEH10 was used to generate gentamicin-resistant Δ*prn* mutant strain. The generation of pEH10 was described previously [29]. Liquid cultures were prepared using Stainer Scholte (SS) medium supplemented with 0.5% (w/v) Heptakis (2,6-di-O-methyl)-β-cyclodextrin) (Sigma H0513). Plate cultures were grown on Bordet Gengou (BG) agar supplemented with 10% (v/v) defibrinated sheep blood (Hemstat, Hemostat Laboratories) and streptomycin (20 μg/mL). Comparative growth curves were generated from triplicate cultures of bacteria grown 48 hours in SS medium at 37°C and shaking at 200 rpm.

## Adherence assay

Adherence assays were conducted following protocols described earlier [12,47]. In brief, human epithelial lung A459 cells were seeded in triplicate in 24-well plates at a density of $2.5 \times 10^5$ cell/well in Dulbecco's modified Eagle's medium (DMEM) (supplemented with 10% fetal bovine serum, 10 mM glutamine, 25 mM sodium pyruvate, 10 mM HEPES). Log-phase bacteria were suspended in warm DMEM medium and added to each well at a multiplicity of infection of 10:1 (bacteria: eukaryotic cells). The plate was centrifuged at 300X*g* for 10 minutes to synchronize infection and the assay plates were incubated for 5 minutes at 37°C. Unattached bacteria were then removed by washing the cells 4 times with 1 mL phosphate-buffered saline (PBS). A459 cells were lysed with 100 μL of 0.1% sodium deoxycholate for 5 minutes and released bacteria suspended in 900 μL of PBS. The bacteria were enumerated by dilution plating on BG agar plates. RB50 Δ*fha*B [12], a mutant strain deleted of the gene encoding the filamentous hemagglutinin and known to be defective in adherence, was used as the negative control.

## Cytotoxicity assay

Cytotoxicity assays were conducted on RAW 264.7 cells, using the CytoTox 96 Nonradioactive Cytotoxicity Assay Kit (Promega) following manufacturer's protocols. In brief, 100 μL of $2.5 \times 10^4$ macrophages were seeded in triplicate in a 96-well plate followed by adding bacteria at a multiplicity of infection of 10:1. The assay plate was centrifuged at 300X$g$ for 10 minutes. Bacteria were incubated with the macrophages for 4 hours, following which the plate was centrifuged for 5 minutes (300X$g$). 50 μL of the supernatant was placed into a fresh flat-bottomed 96-well plate and was calorimetrically assayed for lactate dehydrogenase. A noninfected group was used as a negative control.

## Cytokine test

To assess secretion of TNF-α and IL-1β from RAW 264.7 macrophages, the supernatant of infected cells (MOI = 10 or MOI = 100) was collected at 1-or 4 hpi. To assess the levels of TNF-α and IL-1β in the noses of mice infected with *Bb* WT or *BbΔprn*, noses were collected in PBS and homogenized using a bead tissue disruptor. Cytokine levels were determined using the commercially available ELISA kits R&D Systems Mouse IL-1 beta DuoSet ELISA and TNF-alpha DuoSet ELISA following the manufacturer's instructions.

## Mouse infections

All work with mice was conducted following institutional guidelines. Six-week-old female C57 BL/6 or C3H/HeJ mice (Jackson Laboratories, Bar Harbor, Maine) were used for assessing the colonization and the progress of infection of the respiratory tract. As required, 5–150 CFU of bacteria was delivered in 5 μL of PBS to the nares of mice anesthetized with isoflurane/oxygen. For the colonization profiles, groups of 4 mice were inoculated with WT or mutant bacteria, and at the indicated days 4 mice of each group were euthanized with carbon dioxide ($CO_2$) and the nasal cavity, trachea, and lungs were collected in PBS and homogenized using a bead tissue disruptor. Bacterial load was enumerated by dilution plating.

## Transmission and shedding assay

Transmission assays were conducted using the transmission permissive C3H/HeJ (TLR4 deficient) strain of mouse (Jackson Laboratory) whereby infected (index) mice were co-housed with uninfected (naive) mice [25]. In brief, mice were lightly anaesthetized with isoflurane/oxygen and inoculated with 150 CFU of bacteria delivered in 5 μL of PBS onto the nares. Inoculated (index) mice were then placed in cages with 2 uninfected (naive) mice. Transmission of *B. bronchiseptica* was assessed after 3 weeks of co-housing by enumerating the bacterial load in the nasal cavities of the naive mice. To monitor shedding, the external nares of the index mice were swabbed (32 swipes) with a dry Dacron polyester tipped swab at the indicated times. The swab was vortexed vigorously in 1 mL PBS for 30 seconds and bacteria enumerated on BG agar plates.

## Histopathology

Following fixation in neutral-buffered, 10% formalin solution and subsequent decalcification in Kristensen's solution, coronal sections were made through the nose. Tissues were subsequently processed, embedded in paraffin, sectioned at approximately 5μm, and stained with hematoxylin and eosin. Histopathological examination consisted of evaluation of the nose for the incidence (presence or absence), severity, and distribution of inflammation. Histopathologic severity scores were assigned as grades 0 (no significant histopathological alterations); 1

(minimal); 2 (mild); 3 (moderate); or 4 (severe) based on an increasing extent and/or complexity of change, unless otherwise specified. Lesion distribution was recorded as focal, multifocal, or diffuse, with distribution scores of 1, 2, or 3, respectively [48].

### Flow cytometry

Five mice per experimental group were euthanized by $CO_2$ inhalation. Nasal cavities were harvested, placed in 1mL of RPMI 1640 medium and homogenized via a syringe plunger against a 40μm cell strainer. Cell suspensions were centrifuged at 1,500rpm for 10 minutes and remaining red blood cells were lysed with ACK lysing buffer. After washing with PBS, the cells were incubated with 1μl Zombie aqua (Biolegend) for 20 minutes, washed again, and incubated with 1μl Fc Block (Biolegend) for 30 minutes. Surface marker staining was added to each sample from a master mix of antibodies. Cells were fixed, washed, and resuspended in 250μl FACS buffer. Flow cytometry (Acea Novocyte Quanteon) was performed was used to sort neutrophils (CD11b-Pe-Cy7, CD115-APC, Ly6G-AF488) and macrophages (CD11b-PE-Cy7 CD115-APC, Ly6G-AF488, F4/80-PE). In a separate panel, T cells (CD45-AF700, CD3-APC), B cells (B220-PE-Cy7), and NK cells (NK1.1-PE) were sorted via a separate gating strategy. Viable cells were gated as Zombie aqua-negative cells. The data were analysed with FlowJo 10.0. (S11 Fig).

### Cell isolation for intracellular survival test

The spleen organs harvested from WT or mutant infected Hej mice was cut into 3–4 pieces using sterile scissors. Spleen pieces were mashed with the rough surfaces of two sterile frosted microscope slides and slide surfaces were rinsed with 1 mL (FBS-free) DMEM medium. The samples were then passed through a 70 μm cell strainer to obtain a single-cell suspension. A 100μl sample was plated on BG agar containing 20 μg/ml streptomycin to estimate the number of total bacteria. Cells were washed twice with PBS and then incubated for 1 h with 300 μg/ml gentamicin to eliminate extracellular bacteria. Cells were washed twice with PBS and then lysed with 0.1% triton X-100 for 15 minutes at room temperature. CFU counting was performed on cell lysates by plating 10-fold serial dilutions onto BG agar plates containing 20 μg/ml streptomycin to estimate the number of intracellular bacteria. Control experiments to assess the efficacy of gentamicin were performed in parallel [49]. Briefly, samples of $10^3$, $10^4$ or $10^5$ bacteria were incubated with 300 μg/ml gentamicin for 1 hour at 37˚C and plated on BG agar. There were 3 replicates for each group. The results showed that more than 99.9% bacteria were killed.

### Statistical analysis

Statistical analysis of differences between the WT and mutant groups was performed using the Unpaired Student 2-tailed *t* test, One-way ANOVA and Two-way ANOVA test, as appropriate. P value less than 0.05 shown as *; P value less than 0.01 shown as **; P value less than 0.001 shown as ***; P value less than 0.0001 shown as ****.

### Supporting information

**S1 Fig. Schematic of in-frame deletion of pertactin gene.**
(TIF)

**S2 Fig. Similar laboratory growth *in vitro* of *Bb* WT (blue) and *BbΔprn* (orange) bacteria.**
There were 3 replicates in each time point per group. Error bar shows the standard error of

mean. Statistical significance was calculated using Two-way ANOVA.
(TIF)

**S3 Fig. No effect of pertactin on bacterial adhesion to human alveolar epithelial cells.**
Adherence to A549 lung epithelial cells of *Bb* WT (blue), *BbΔprn* (orange) and *BbΔfha* (yellow) that was previously shown to be impaired in its ability to adhere to epithelial cells. There were 3 replicates in each group. Error bar shows the standard error of mean. Statistical significance was calculated using One-way ANOVA. ***$p < 0.001$.
(TIF)

**S4 Fig. No significant difference in cell cytotoxicity of *Bb* WT and *BbΔprn* to RAW 264.7 macrophages.** There were 3 replicates in each group. Error bar shows the standard error of mean. Statistical significance was calculated using One-way ANOVA. ****$p < 0.0001$.
(TIF)

**S5 Fig. Comparative colonization profiles of *Bb* WT and *BbΔprn* in C57 BL/6 mice.** Bacterial CFU recovered on days 3-, 7-, 14-, and 28 dpi from the nasal cavities, trachea, and lungs of mice inoculated with either wild-type (blue) or mutant (orange) bacteria. There were 4 mice in each time point per group. Error bar shows the standard error of mean. Statistical significance was calculated by Two-way ANOVA. *$p < 0.05$, **$p < 0.01$, ***$p < 0.001$.
(TIF)

**S6 Fig. The role of PRN in colonization of *B. bronchiseptica* in C3H/HeJ mice.** Comparative colonization profiles of *Bb* WT and *BbΔprn* in C3H/HeJ mice. Number of colony-forming units (CFU) recovered on days 3-, 7-, 14-, and 28 pi from the nasal cavities, trachea, and lungs of mice infected with either wild-type (blue) or mutant (orange) bacteria. There were 4 mice in each time point per group. Error bar shows the standard error of mean. Statistical significance was calculated by using Two-way ANOVA.
(TIF)

**S7 Fig. *B. bronchiseptica* PRN is not required for efficient colonization.** $ID_{50}$ test of *Bb* WT and *BbΔprn* showing bacterial numbers at 7 dpi in respiratory organs of C3H/HeJ mice inoculated with incrementally increasing doses of 5 CFU, 25 CFU and 125 CFU. Number of CFU recovered from nasal cavities, trachea, and lungs revealed no difference in colonization between wild-type and mutant. There were 4 mice in each time point per group. Error bar shows the standard error of mean. Statistical significance was calculated by using Unpaired t-test.
(TIF)

**S8 Fig. TNF-α and IL-1β levels in noses of mice infected with *Bb* WT or *BbΔprn* at 24hpi or 48hpi.** The levels of TNF-α in noses of mice infected with *Bb* WT (blue) or *BbΔprn* (orange) at 24hpi (A) or 48hpi (C). The levels of IL-1β in noses of mice infected with *Bb* WT (blue) or *BbΔprn* (orange) at 24hpi (B) or 48hpi (D). Error bar shows the standard error of mean. Statistical significance was calculated using unpaired T test.
(TIF)

**S9 Fig. PRN does not induce the secretion of IL-1β from RAW 264.7 macrophages.** A) The IL-1β detected in the supernatant when RAW macrophages were exposed with *Bb* WT (blue) or *BbΔprn* (orange) with a MOI at 10 for 1 hour. B) The IL-1β detected in the supernatant when RAW macrophages were exposed with *Bb* WT (blue) or *BbΔprn* (orange) with a MOI at 100 for 1 hour. Error bar shows the standard error of mean. Statistical significance was calculated using unpaired T test.
(TIF)

**S10 Fig. PRN contributed to the generation of anti-*B. bronchiseptica* IgG antibodies.** IgG antibody titers against *B. bronchiseptica* were determined in sera of C3H/HeJ mice infected with either *Bb* WT (blue) or *BbΔprn* (orange) at 28 dpi. There were 4 mice per group. Error bars show the standard error of mean. Statistical significance was calculated by using One-way ANOVA. $^*p < 0.05$, $^{**}p < 0.01$, $^{***}p < 0.001$.
(TIF)

**S11 Fig. Gating strategy for flow cytometry.** A) Gating strategy for myeloid cell types. B) Gating strategy for lymphoid cell types.
(TIF)

**S1 Table. Scoring standard for histopathology analysis.**
(TIF)

## Acknowledgments

We acknowledge the staff of the mouse facilities at the University Research Animal Resources (URAR), University of Georgia, for facilitating the work. We also thank members of the Harvill laboratory for helpful discussions and editing of the manuscript.

## Author Contributions

**Conceptualization:** Eric T. Harvill.

**Data curation:** Longhuan Ma.

**Formal analysis:** Longhuan Ma, Kalyan K. Dewan, Dawn L. Taylor-Mulneix, Shannon M. Wagner, Bodo Linz, Israel Rivera, Yang Su, Uriel Blas-Machado.

**Funding acquisition:** Eric T. Harvill.

**Investigation:** Longhuan Ma, Kalyan K. Dewan, Dawn L. Taylor-Mulneix, Shannon M. Wagner, Bodo Linz, Israel Rivera, Yang Su, Amanda D. Caulfield, Uriel Blas-Machado, Eric T. Harvill.

**Methodology:** Longhuan Ma, Kalyan K. Dewan, Dawn L. Taylor-Mulneix, Shannon M. Wagner, Bodo Linz, Israel Rivera, Eric T. Harvill.

**Supervision:** Eric T. Harvill.

**Validation:** Longhuan Ma.

**Writing – original draft:** Longhuan Ma, Eric T. Harvill.

**Writing – review & editing:** Longhuan Ma, Kalyan K. Dewan, Bodo Linz, Amanda D. Caulfield, Uriel Blas-Machado, Eric T. Harvill.

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
