## [Decision Letter · Decision Letter 0]

8 Dec 2020

Dear Mr. Ma,

Thank you very much for submitting your manuscript "Pertactin contributes to shedding and transmission of Bordetella bronchiseptica" for consideration at PLOS Pathogens. As with all papers reviewed by the journal, your manuscript was reviewed by members of the editorial board and by several independent reviewers. In light of the reviews (below this email), we would like to invite the resubmission of a significantly-revised version that takes into account the reviewers' comments.

While this manuscript does provide some novel findings the manuscript as it currently stands does not reach the threshold for publication in PlosPathogen. Increased mechanistic insight is required to better prove the role of Prn and most importantly a better attempt must be made to translate the relevance of the model to B. pertussis infection in humans.

We cannot make any decision about publication until we have seen the revised manuscript and your response to the reviewers' comments. Your revised manuscript is also likely to be sent to reviewers for further evaluation.

Sincerely,

Rachel M McLoughlin, PhD

Associate Editor

PLOS Pathogens

David Skurnik

Section Editor

PLOS Pathogens

Kasturi Haldar

Editor-in-Chief

PLOS Pathogens

orcid.org/0000-0001-5065-158X

Michael Malim

Editor-in-Chief

PLOS Pathogens

orcid.org/0000-0002-7699-2064

While this manuscript does provide some novel findings the manuscript as it currently stands does not reach the threshold for publication in PlosPathogen. Increased mechanistic insight is required to better prove the role of Prn and most importantly a better attempt must be made to translate the relevance of the model to B. pertussis infection in humans.

Reviewer's Responses to Questions

**Part I - Summary**

Reviewer #1: There are some novel and potentially interesting findings in this study, but unfortunately it is very thin, lacking detail and mechanistic insight. The authors need to do a good deal more work before this story is complete.

Reviewer #2: The paper addresses a long standing query in the Bordetella field: the true role of Prn in infection biology. Early studies suggested Prn to be an adhesin, and the presence of an RGD motif supports this. However, clear evidence for this has been obtained from in vitro studies only, and in vivo studies demonstrated efficient colonisation in the absence of Prn.

Here, careful studies using very small inoculae very clearly demonstrate that BB prn mutants can very efficiently colonise and persist in the respiratory tracts of mice, demonstrating that Prn is not required for these processes in BB.

Instead, data is presented to argue for a role in intracellular survival, and in inducing inflammatory responses to BB infection. Of particular interest is the demonstration of reduced shedding of BB prn mutants from the respiratory tract and reduced transmission between index and recipient mice. The authors combine these findings to suggest a role for Prn in inducing inflammation, that facilitates bacterial shedding and onward transmission.

This story makes a telling contribution to defining the infection biology of the bordetellae, and adds to understanding the role of Prn in this. Prn is of particular importance as it is one of the components of many of the acellular Pertussis vaccines. Importantly, in countries experiencing resurgence of pertussis, there is a stark increase in the frequency of isolates that are Prn-deficient, and there are multiple lines of evidence to suggest that Prn-deficient B. pertussis strains have a fitness advantage in populations that are highly vaccinated with the acellular vaccines.

In the light of this, the study is important, as it addresses the role of pertactin in infection which is crucial to understanding the consequences of Prn deficiency for B. pertussis fitness.

Reviewer #3: Using a TLR4 deficient mouse model, Ma et al explore the role of Pertactin (PRN) of B. bronchiseptica in colonization, growth, spread within the respiratory tract, shedding, and transmission. It was demonstrated that while PRN deletion mutants showed no change in colonization of the trachea and lungs of mice compared to the WT strain, shedding and transmission was significantly reduced. The respiratory epithelia of mice infected with the PRN deletion mutant exhibited significantly reduced inflammation and mucus accumulation compared to the WT strain. Although the results suggest an important role for pertactin in B. bronchiseptica, the authors overstate what their model shows with regards to B. pertussis infections.

The argument that studying the B. bronchiseptica in one of its natural hosts provides valuable insights into B. pertussis infections in humans would only be valid if the diseases caused by the two pathogens in their natural hosts were comparable. Although the bacteria are genetically similar and share some virulence factors, the diseases caused by B. bronchiseptica and B. pertussis are very different, with different tropisms (humans/NHP vs. many mammals except humans) and clinical presentations (acute/weeks vs prolonged/chronic). Additionally, pertussis toxin is a key toxin and essential virulence factor in pertussis infections (PT-only vaccines prevent disease), yet it is not expressed by B. bronchiseptica—highlighting a major difference in disease mechanisms. Bb and Bp also express different lipid A structures and polysaccharide chains (LPS vs LOS), with Bb having a more inflammatory lipid A/LPS (2). Using a TLR4-deficient mouse strain to allow disease transmission further confounds the applicability of these results to B. pertussis infections, since TLR4 appears to have different roles in Bb and Bp infections (3). PRN’s contribution to inflammation, which the researchers suggest may influence mucus production, shedding and transmission, may be insignificant in a TLR4-competent mouse (a key innate immune recognition molecule for endotoxin and host DAMPs). It is possible the effects observed are artifacts of the experimental model.

B. pertussis survives very poorly outside the host whereas B. bronchiseptica has been shown to survive for prolonged periods of time in water. B bronchiseptica can be efficiently transferred by contact and by sharing of water sources. This is not true of B pertussis which is transmitted by respiratory droplets and aerosols.

**Part II – Major Issues: Key Experiments Required for Acceptance**

Reviewer #1: 1) The authors justify their research on pertactin (PRN) from Bordetella on the basis of the observations that pertactin-deleted mutants of Bordetella pertussis have emerged in populations immunized with acellular pertussis vaccine that include PRN as an antigen. However, they carry out their research with B. bronchiseptica and the justification is that they have a mouse transmission model with this bacteria, which they don’t have with B. pertussis. However, many aspects of the study, local inflammation, mucus secretion and intracellular survival could have been done with B. pertussis, where they could have used human clinical isolates, rather than lab-generated mutants of B. bronchiseptica.

2) The biggest issue with this report is that it lacks depth, the essential data could be condensed into 2 figures. Do we need a whole figure showing only two bars on a bar graph (Fig. 3) These bars don’t have error bars, making less of case for including them in a figure at all. The first two figures of negative data (Fig 1 and 2) that could be one supplementary figure. The 3 interesting observations are a) transmission and shedding is lower in mice infected with the PRN-neg mutant, b) there is less inflammation/mucus secretion in nasal cavity of mice infected with the PRN-neg mutant c) intracellular survival in macrophages in vitro is lower with the PRN-neg mutant. However, none of these observations are linked. The authors speculate on immune responses being stronger in mice infected with WT compared with PRN-negative mutants of B. bronchiseptica, but have not examined this. They speculate about higher inflammasome activation with WT bacteria, but have not assessed this. There is not data linking either intracellular survival or inflammation with transmission.

3) The authors should consider some studies with purified PRN where they look at its effect on inflammatory responses of macrophages in vitro. There is some evidence that PRN works with FHA to suppress LPS-induced inflammatory responses by macrophages. It is not clear how those observation would fit with the suggestion in the current study that PRN may promote inflammation. Either way, it would worth testing the effect of PRN on induction or inhibition of inflammasome activation? Studies such as these might help to expand on their theories around the inflammasome.

4) The B. pertussis challenge dose for the early figures is 0.5 million CFU, but is only 150 CFU in the later studies, how can these be reconciled or compared in making conclusion on the course of infection versus inflammation or shedding etc.

Reviewer #2: A key message is that Prn contributes to the induction of inflammation by BB. As it stands, it is not possible to determine if the data supports this.

Ln342-351. More detail is required for the approach to histopathology scoring. Currently, it is not possible to understand how scores were determined. For example, grade 0- no significant alterations. What counts as significant? Grade 1 – minimal, which would be zero? Scoring based partly on ‘complexity’ of change. What does this mean? What area of the total tissue was scored? How was bias excluded from the selection of areas. Were observers blinded to the source in terms of WT or mutant? I appreciate that drawing quantitative measures from these sorts of data is difficult, but there needs to be greater clarity as to how quantitative measures were drawn from visual images for the conclusion regarding the role of Prn in inflammation to stand up to scrutiny.

Reviewer #3: Comments:

1. The authors state that RB50 transmits efficiently between mice (lines 143-144) but that only appears to be the case in an immuno-deficient mouse strain. In order to draw conclusions about the transmission of PRN-negative B. bronchiseptica strains, the authors should explore animal models of B. bronchiseptica transmission that don’t require the use of immunodeficient animals (e.g. the pig model)

2. The researchers should complement or repair the isogenic PRN KO, to confirm that unintentional, off-target effects are not contributing to their disease models (mice, cell culture).

3. The researchers should consider assessing inflammasome activation in their in vitro intracellular invasion system as they speculate that may explain differences in inflammation induced by the WT and PRN-deficient strains.

**Part III – Minor Issues: Editorial and Data Presentation Modifications**

Reviewer #1: The preparation of manuscript, especially the figures, are not up to the standard required for a top-class journal. e.g. out of sequence figure call out, references to figures in the Introduction and discussion, poor quality figures; it is almost impossible to read the font on some figures. The Y axis scales on panels within figures are not consistent. Some Y axes are split when they don’t need to be, and many don’t start at zero.

I think many people in the field would disagree with the statement in the introduction that whole cell vaccine had “relatively minor side effects”.

Reviewer #2: There is a clear contradiction between Prn contributing to transmission and the striking increase in Prn-deficiency among B. pertussis. The authors mention possible compensation for loss of Prn from B. pertussis by other autotransporters, but there is no clear rationale for this. Could the authors comment on the difference between BB and B. pertussis in the ris system that has been linked to intracellular survival, and for which Chen and Stibbitz suggested a role in promoting the expression of Vrgs and aerosol transmission (Curr Opin Microbiol. 2019. DOI: 10.1016/j.mib.2019.01.002. The authors should also acknowledge that their findings could indicate that Prn has a different role in B. pertussis compared to BB. This does not question the findings reported here regarding the role for Prn in BB infection, but if a key motivation for this study was to address the question of Prn and Prn-deficiency among B. pertussis, then a more rounded discussion of this is required.

There are a number of other points that the authors could consider:

Ln 117. Describing BB and BP as 98% identical at the nucleotide level is misleading. Speciation of B. pertussis involved loss of over 1Mb of DNA compared to BB, describing identity as 98% infers there is only 2% difference between the two.

Ln119. ‘ref5’ suggests a citation issue.

Ln131. It is unusual to refer to results figures in the introductiomn.

Ln209-212. This is discussion not results.

Ln204. States RAW cells were infected with bacteria at an MOI of 10 and 1, materials states 100, 10 and 1. States extracellular bacteria were killed by 1 and 2 hours of gent treatment. But in Ln276-8, it suggests Prn mutant is gent resistant? Ln 315 states extracellular bacteria were killed with polymyxin but this was added 1 hour after addition of bacteria to the RAWs. There appears contradiction between these sections.

If polymyxin B was used, is the sensitivity of WT and the Prn mutant to PMB identical, particularly at low levels? Some PMB will cross the RAW membranes and an increased sensitivity of the Prn mutant would reduce numbers recovered. The cytotoxicity of BB for mammalian cells could compromise cell membranes. The same applies for Triton X-100 as this was used to lyse the RAW cells to recover the intracellular bacteria. It is important to show that different sensitivity was not the reason for the difference in recovered bacteria.

Figure 1. For transmission, index mice were co-housed with naïve mice for 21 days. Wt mice were cleared from the lungs of the mice sometime between the day 14 and day 28 time points. It is not clear when these WT were cleared, it might have been day 27, or day 15. Prn mutants were recovered from the lungs on day 28. Thus the colonisation profiles of WT and prn mutants are different, although not in the region where shedding would be expected to occur from (nasal cavity), and this wouldn’t explain reduced shedding of Prn mutants, but it’s not correct to say no difference.

Figure 2. It is very difficult to be precise in getting exactly 5 cfu per inoculum. Were the inoculae plated? If so, show data to show the likely range of inoculate the mice will have received for each of the 3 categories. The data suggest that none of the mice received zero cfu?

Ln189, In the text, Figure 5A and B are identified as showing histopathology induced by WT BB, but in the Fig, A-C are panels showing sections from control mice. This might stem from the odd numbering for Figure 5 which appears to be 3 quite separate figures, but are included in one Figure labelled at 5A-C. However, 5A then has panels A-I, so creates an odd labelling of 5AA, 5AB but this is not used in the text.

Fig 5A. It is notoriously difficult to demonstrate a clear difference between pathology using single panels from single samples. As mentioned above, some explanation of how histopathology was scored, and how cell sections were selected for analysis in an unbiased way. It is difficult for the reader to understand the data depicted in Figs 5B and C without this.

Figure 6. For the intracellular survival of BB in RAWs, the data is presented as a survival ratio, but there is no explanation of what this is, or how it was calculated. In Fig 6A data is shown for WT and Prn mutants at MOIs of 1 and 10 for 2 hours post internalisation. In Fig 6B data is shown for 1, 2, 4 and 24 hours post internalisation, but the data for 2 hours appear to be different to that in Fig 6A. Are these data from a different experiment, or a different MOI?

Reviewer #3: 1. Researchers did not indicate if their p-values for student T tests were corrected for multiple comparisons, which may impact the statistical validity of the results. It is possible that more mice are needed to reach statistical significance, and indeed may allow for some statistical comparisons between WT and PRN-KO mice in terms of dissemination to the spleen at day 7 and 14. Animal numbers and error bars (SD? SEM?) should be described in the figure legends.

2. The statement that natural B. pertussis infections begin in the nasopharyngeal region, progress slowly during a prolonged catarrhal stage and do not normally even reach the lungs (lines 63-65). Should be deleted or supported by primary references. The natural progression of B. pertussis colonization in humans is unknown and has not been studied. The data that’s available from human autopsy cases and NHP studies suggests B. pertussis penetrates deeply into the airway.

3. Line 235: “Bb WT inoculated mice shed significant more bacteria than BbΔprn inoculated mice” cite Fig. 4

2. Line 247-249: “In addition, Bb WT showed significantly higher intracellular survival in macrophages compared with the PRN deletion mutant in an in vitro assay” cite Fig. 6

3. Line 255-257: “Thus, increased intracellular survival may result in a stronger immune response against Bb WT than BbΔprn, consistent with the histopathology analysis in this study” cite Fig. 5

PLOS authors have the option to publish the peer review history of their article (what does this mean?). If published, this will include your full peer review and any attached files.

Reviewer #1: No

Reviewer #2: No

Reviewer #3: No
---

## [Decision Letter · Decision Letter 1]

22 Mar 2021

Dear Mr. Ma,

Thank you very much for submitting your manuscript "Pertactin contributes to shedding and transmission of Bordetella bronchiseptica" for consideration at PLOS Pathogens. As with all papers reviewed by the journal, your manuscript was reviewed by members of the editorial board and by several independent reviewers. 

While there is overall enthusiasm for this study there still remains a number of outstanding issues. These will need to be addressed in full prior to publication being considered.

In particular, the lack of mechanistic data to explain the role of PRN in inflammation. The reviewers had suggested assessing immune responses and inflammasome activation induced by WT versus PRN- mutant bacteria to explain the higher inflammation with the WT bacteria and although you have looked at immune cell recruitment in the respiratory tissue, which is useful, this is very superficial and not a readout of immune response. The inflammasome activation theory has not been supported by any data. Indeed, the new data suggests that the mutant bacteria induce more, not less, IL-18 production from macrophages which you suggest may be due to the cell type chosen. If this is the case then perhaps an alternative cell type should be considered.

Secondly the lack of data with purified PRN. In your response you mention an experiment to look at inflammasome activation via IL-1 and IL-18 production in the reply to reviewers but show no data because the experiment didn’t seem to work. This is may be because two signals for inflammasome activation have not been used. It will be important to definitively prove or disprove the theory on inflammasome activation.

Overall the insights into the biological action of pertactin are interesting but it's difficult to draw conclusions about the role of pertactin in B. pertussis infections since the rapid loss of pertactin from circulating strains indicates it is not required for infection or transmission of B. pertussis within the human population.

Finally, while it is agreed that models do not have to recreate every aspect of a disease system to be useful, but in order for a model to be useful, it’s strengths and weaknesses have to be considered and discussed. In the introduction for example the significant differences between B. bronchiseptica and B. pertussis infections are somewhat glossed over while emphasizing the similarities between the two strain.

We cannot make any decision about publication until we have seen the revised manuscript and your response to these comments. Your revised manuscript is also likely to be sent to reviewers for further evaluation.

Sincerely,

Rachel M McLoughlin, PhD

Associate Editor

PLOS Pathogens

David Skurnik

Section Editor

PLOS Pathogens

Kasturi Haldar

Editor-in-Chief

PLOS Pathogens

orcid.org/0000-0001-5065-158X

Michael Malim

Editor-in-Chief

PLOS Pathogens

orcid.org/0000-0002-7699-2064

Dear Authors

While there is overall enthusiasm for this study there still remains a number of outstanding issues. These will need to be addressed in full prior to publication being considered.

In particular, the lack of mechanistic data to explain the role of PRN in inflammation. The reviewers had suggested assessing immune responses and inflammasome activation induced by WT versus PRN- mutant bacteria to explain the higher inflammation with the WT bacteria and although you have looked at immune cell recruitment in the respiratory tissue, which is useful, this is very superficial and not a readout of immune response. The inflammasome activation theory has not been supported by any data. Indeed, the new data suggests that the mutant bacteria induce more, not less, IL-18 production from macrophages which you suggest may be due to the cell type chosen. If this is the case then perhaps an alternative cell type should be considered.

Secondly the lack of data with purified PRN. In your response you mention an experiment to look at inflammasome activation via IL-1 and IL-18 production in the reply to reviewers but show no data because the experiment didn’t seem to work. This is may be because two signals for inflammasome activation have not been used. It will be important to definitively prove or disprove the theory on inflammasome activation.

Overall the insights into the biological action of pertactin are interesting but it's difficult to draw conclusions about the role of pertactin in B. pertussis infections since the rapid loss of pertactin from circulating strains indicates it is not required for infection or transmission of B. pertussis within the human population.

Finally, while it is agreed that models do not have to recreate every aspect of a disease system to be useful, but in order for a model to be useful, it’s strengths and weaknesses have to be considered and discussed. In the introduction for example the significant differences between B. bronchiseptica and B. pertussis infections are somewhat glossed over while emphasizing the similarities between the two strain.
---

## [Editor Report · Decision Letter 2]

21 Jun 2021

Dear Mr. Ma,

We are pleased to inform you that your manuscript 'Pertactin contributes to shedding and transmission of Bordetella bronchiseptica' has been provisionally accepted for publication in PLOS Pathogens.

Best regards,

Rachel M McLoughlin, PhD

Associate Editor

PLOS Pathogens

David Skurnik

Section Editor

PLOS Pathogens

Kasturi Haldar

Editor-in-Chief

PLOS Pathogens

orcid.org/0000-0001-5065-158X

Michael Malim

Editor-in-Chief

PLOS Pathogens

orcid.org/0000-0002-7699-2064
---

## [Editor Report · Acceptance letter]

20 Jul 2021

Dear Mr. Ma,

We are delighted to inform you that your manuscript, "Pertactin contributes to shedding and transmission of *Bordetella bronchiseptica*," has been formally accepted for publication in PLOS Pathogens.

Best regards,

Kasturi Haldar

Editor-in-Chief

PLOS Pathogens

orcid.org/0000-0001-5065-158X

Michael Malim

Editor-in-Chief

PLOS Pathogens

orcid.org/0000-0002-7699-2064